



# tTEM20AAR: a benchmark geophysical dataset for unconsolidated fluvio-glacial sediments

Alexis Neven[1], Pradip Kumar Maurya[2], Anders Vest Christiansen[2], and Philippe Renard[1,3]

[1]Centre of Hydrogeology and Geothermics, University of Neuchâtel, Neuchâtel, Switzerland
[2]Department of Earth Sciences, Aarhus University, Aarhus C, Denmark
[3]Department of Geosciences, University of Oslo, Oslo, Norway

**Correspondence:** Alexis Neven (alexis.neven@unine.ch)

**Abstract.** Quaternary deposits are complex and heterogeneous. They contain some of the most abundant and extensively used aquifers. In order to improve the knowledge of the spatial heterogeneity of such deposits, we acquired a large (more than 1400 hectares) and dense (20 m spacing) Time Domain ElectroMagnetic (TDEM) dataset in the upper Aare Valley, Switzerland. TDEM is a fast and reliable method to measure the magnetic field directly related to the resistivity of the underground. In this
paper, we present the inverted resistivity models derived from this acquisition, and all the necessary data in order to perform different inversions on the processed data (https://doi.org/10.5281/ZENODO.4269887 (Neven et al., 2020)). The depth of investigation ranges between 40 to 120 m depth, with an average data residual contained in the standard deviation of the data. These data can be used for many different purposes: from sedimentological interpretation of quaternary environments in alpine environments, geological and hydrogeological modeling, to benchmarking geophysical inversion techniques.

## 1   Introduction

In most urbanised and agricultural areas of Switzerland, the shallow underground is constituted of Quaternary deposits. The thickness can vary from few meters to hundreds of meters. These recent sediments are deposited by various agents such as rivers, lakes, glaciers or even landslides. Each time, the associated sediment will have a different composition, permeability, and spatial extents. Furthermore, they might be all intertwined (a lake deposit can be partially eroded by a glacier, and refilled
with river sediments). This leads to a spatial variability that is often higher than expected in such deposits.

These formations cover almost 30% of Switzerland, and includes all the most populated areas. In addition, these formations are some of the most solicited: water supply for cities, extraction of construction materials, geotechnical constructions and shallow geothermal exploitation. Often, the construction of geological models using only boreholes can miss most of the spatial heterogeneity, and conduct to inadequate models and wrong conclusions. Increasing the number of boreholes to reduce
the uncertainty is often difficult and expensive. A good example of these highly exploited Quaternary zones is the upper Aare Valley. In the 20 km by 3 km rectangle defined by the valley side limits (Fig. 1), the city of Thun and the city of Bern, the Aare Valley includes 4 quarries, 6350 pumping wells (Shallow geothermic or drinkable water) and 5300 injection wells (re-inject water after geothermal heat pump). A previous valley size model was designed using boreholes and surface data (Volken et al., 2016), but the model does not represent the internal heterogeneities of the Quaternary formations and can show unrealistic



sharp variations due to the nearest neighbors interpolation method used during the workflow. Therefore, there is a need for a better understanding of Quaternary sedimentary heterogeneity, in order to better constrain geological models, knowledge that could be applied in the Aare valley or for any fluvio-glacial filling area.

    Near-surface geophysics such as DC resistivity, electromagnetic or seismic methods can bring important information in terms of the spatial distribution of facies. However, they are usually carried out in restricted areas to answer specific local questions,

and do not help to understand the variations of geology at the valley scale. Consequently with such datasets, it is difficult to develop general modeling methods that can be applied to fluvio-glacial deposits, or even to come up with simple conceptual models for large scale filling. In order to fill this gap of information, and provide a valley scale fluvio-glacial resistivity map, we conducted in January 2020 a large geophysical survey using tTEM (towed Transient Electromagnetic) system(Auken et al., 2019) in the upper Aare Valley, Switzerland. The tTEM-system provides a very detailed (both vertically and horizontally)

resistivity model. The *tTEM20AAR* dataset covers a section of the valley of approximately 1-2 km width and 16 km long. The fields were mapped with a line spacing of 20 meters, resulting in about 1500 hectares of covered land (see Fig. 1). The raw tTEM data were processed to suppress and removed noisy data parts, and then inverted to a resistivity model using specially constrained inversion algorithm(Viezzoli et al., 2008). The resulting resistivity model consists of 57'000 1D models of 30 layers. The depth of investigation varies, from 40 to 120 meters depth, primary driven by lithological/resistivity variations. The

resulting resistivity model explains (fits) the recorded data well within the estimated data uncertainty. The resistivity model reveals new and very interesting geophysical/geological structures of the subsurface at a fine resolution.

    The *tTEM20AAR* data set can be used for several purposes. It can be used as a benchmark to test and compare geophysical inversion procedures for tTEM systems. For example, in this work the inversion was carried out in 1D for every vertical sounding but using constraints to account for lateral continuity. The full 3D inversion is much more demanding, but the data

set could be used to compare it with the results published in this paper. Stochastic inversion (Mosegaard and Tarantola, 1995; Linde et al., 2017) using different methods or types of prior knowledge could also be applied on this data set and be compared with the published results. More generally, Quaternary formations are highly heterogeneous and constitute a challenge for geostatistical and uncertainty modeling (De Marsily et al., 2005). Sharing this data set will allow to test and compare various methods to interpolate the properties of the underground and construct models that can be used for various purposes. The

integration of geophysical methods to constrain hydrogeological models is also a very important field of research (Binley et al., 2015). The *tTEM20AAR* data set may help testing the development of innovative methods for the construction of groundwater models. It is important to note in this perspective that the Upper Aare Valley has been extensively studied and a consequent amount of additional data are distributed by the Swiss authorities : a few thousand of logged boreholes, 46 stations of present or historic groundwater level measurements, a complete bedrock elevation model and Quaternary deposits thickness, as well

as a complete hydrological and geological map. All these data cover the same site, and in combination with the geophysical data presented in this paper, it forms an ideal test site for multiple data integration. Especially, we consider this site ideal for benchmarking new methods that integrates hydrological and geophysical data, or more generally that may link porosity, permeability or lithology with resistivity. Improvement in such methods may strongly improve hydrogeological modeling in such environments, subject to high local facies variations. Finally, from a more geological perspective, the *tTEM20AAR* data

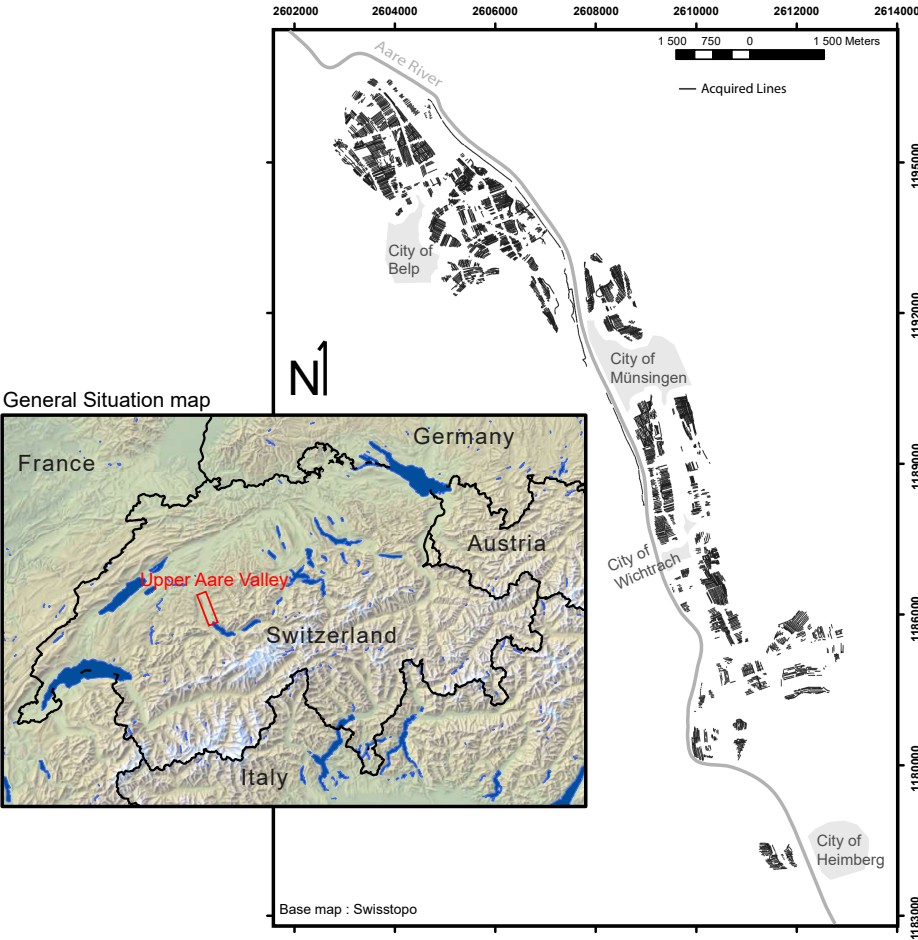

**Figure 1.** Location of the study area and acquisition lines. Coordinates are in meters in the UTM 32N reference. Base map from the Swiss Topographic office.

set could be used to better understand the internal structures of Quaternary deposits within alpine valleys. It could be analysed in detail from a sedimentological perspective and used to better constrain the glacial and geological history of the Quaternary deposits (Preusser et al., 2011).

## 2   Methods

### 2.1   The tTEM-system

The tTEM-system used for the data acquisition is developed by the HGG-group at Aarhus University, Denmark (Auken et al., 2019). The tTEM-system is a towed, ground-based, transient electromagnetic system, designed for high efficient data collection and detailed 3D-mapping of the shallow subsurface (the upper  80 m). TEM-methods build on the principle of induction





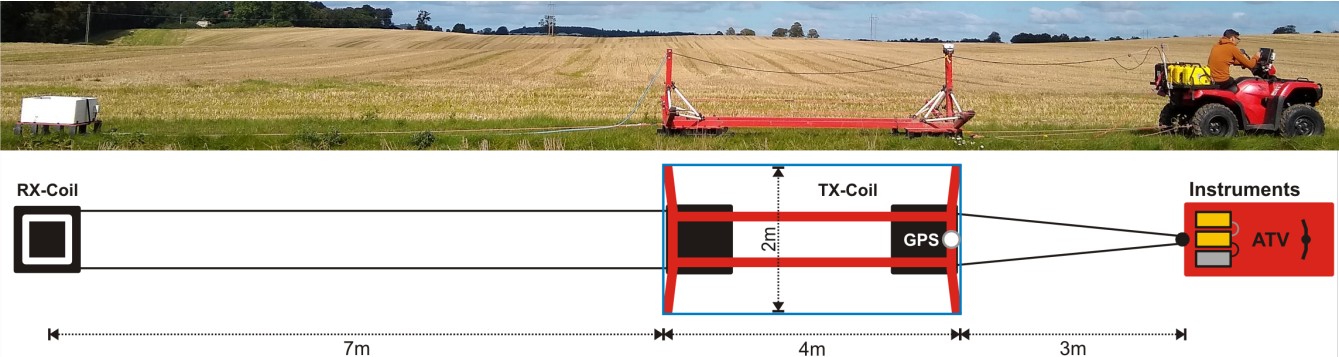

**Figure 2.** The tTEM system.

(Faraday's law of induction) for mapping the electrical conductivity (conductivity=1/resistivity) of the subsurface. A detailed description of the TEM principle can be found in (Christiansen et al.). The layout of the tTEM system is shown in Fig. 2.

The tTEM-system consists of an All-Terrain Vehicle (ATV) carrying the instrumentation and towing the transmitter frame (Tx coil) and the receiver coil (Rx coil) in an offset configuration. The Tx and Rx coils are mounted on sleds for all terrain capability. All frame parts and sleds are built of non-conductive composite materials. Driving path and various data quality control parameters are monitored in real time by the driver on a mounted screen. Operation speed is up to 20 km/h. We used an off-set configuration, where the receiver coil is 7 meters behind the transmitter coil. Both of them are horizontal, allowing

to measure the $z$ component of the secondary magnetic field. A GPS is mounted on the frame to ensure correct positioning of the data. The transmitter loop consists of one loop of 4x2 m, creating an area of 8 m$^2$. We used a standard dual moment TEM configuration: a high moment (HM) with a high inductive current of 30 A and a low moment with a lower inductive current of 5 A. Such configuration has the advantage of being able to resolve shallow targets with the low moment and its associated fast turn off time, and to reach higher penetration depth with the high moment. Both moments are stacked few hundreds times.

Detailed parameters are summarized in the table 1. The gate is the time interval in which the received amplitudes are averaged. Due to the signal attenuation, further we get in the listening time, lower is the signal to noise ratio. In order to partially counterbalance this effect, we used a logarithmic increasing gate size related to listening time.

To ensure the data quality, the tTEM-instrumentation were calibrated prior to the survey at the Danish national TEM test site following the calibration procedure described by Foged et al. (2013). The two calibrated parameters are a time shift and an

amplitude factor. The calibration was done with the ATV connected to the equipment in order to account for any shift caused by it. Figure 3 shows the match between the test site reference response and the measured tTEM-response after calibration, which results in a fully acceptable match.

## 2.2    Field Site

The field site is the Upper Aare Valley, in central Switzerland (see Fig. 1). The survey took place in January 2020. During

approximately 15 working days, we covered all the accessible farming fields in the valley along a 26 km long section. The

**Table 1.** Specifications of the High and Low moment used in the acquisition. The gate size increases with time in order to counterbalance less good signal to noise ratio due to the wave attenuation.

| Parameters | LM | HM |
|---|---|---|
| No. of turns | 1 | |
| Tx coil area | 8 m$^2$ | |
| Transmitter current | 5 A | 30 A |
| Peak moment | 30 Am$^2$ | 240 Am$^2$ |
| Repetition frequency | 1055 Hz | 315 Hz |
| Stacks | 422 | 252 |
| Total cyclus time | 0.22 s | 0.40 s |
| Tx time | 0.2 ms | 0.45 ms |
| Turn off time | 2.8 µs | 4.5 µs |
| Number of gates | 4 | 23 |
| Gate size | 4 µs - 10 µs | 10 µs - 900 µs |
| First gate start | 4.38 µs | 10.30 µs |

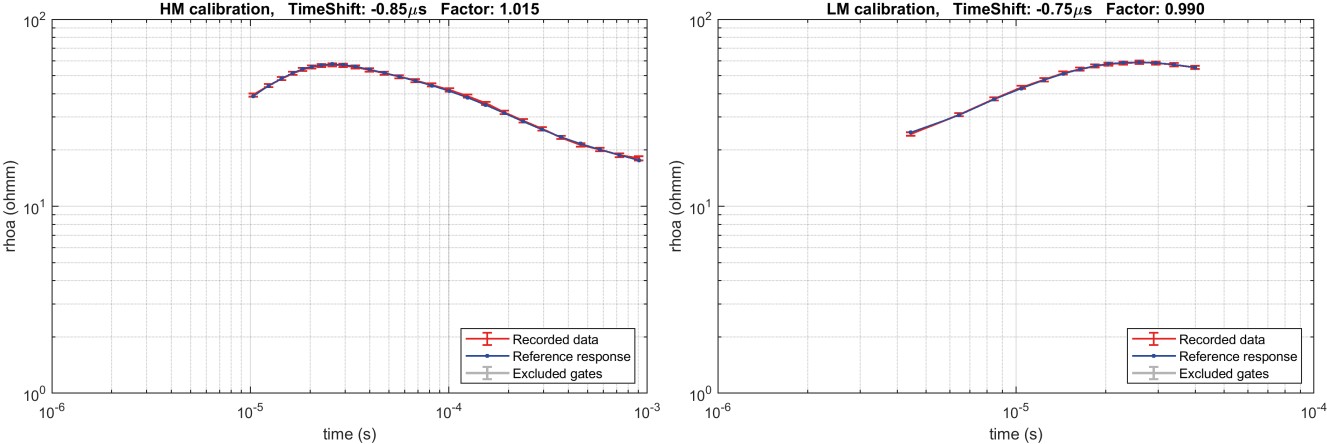

**Figure 3.** Calibration of the High and Low moment. The apparent resistivity (rhoa) is compared with the reference response). No gates were excluded. The resulting time shift and scale factor are respectively -0.75 µs and 0.99 for the LM, and - 0.85 µs and 1.015 for the HM.

driving speed was between 10 to 20 km/h, depending of the terrain. Since the acquisition rate is time dependant, and not distance triggered, we also lowered the speed in noisier or less responsive areas in order to acquire a denser dataset. The spacing between the lines was approximately 20 meters. The average covered surface par day was 112 hectares, for a total of 1425 hectares.



## 2.3 Data Processing

The voltage data from the receiver is measured continuously, and need to be cleaned of man-made noise and coupling. Data processing and inversion were carried out with the tTEM processing module in the *Aarhus Workbench* software. The objective of the processing of the tTEM-data is to remove any interference in the data from man-made installation (coupled data), suppress random noise by stacking, and finally discard the noisy late time data entering the background noise. Thus, we

ensure that the resulting resistivity model represents geological structures of the subsurface without artifact from man-made installation. Processing of the dB/dt data comprises of the following steps, which is a standard EM processing workflow :

- Detection of coupled structures in the data and filtering.

- Averaging of raw data to suppress random noise. Raw data are averaged using a moving average filter with narrow time windows in early times and wider in the late times.

- Creation of vertical soundings every 2.5 s which corresponds approximately to a spacing of 10 m. The exact distance can vary depending on driving speed.

- Automatic filtering of the averaged data for removal of late-time data points entering the background noise.

- Visual assessment of all dB/dt data and manual removal of coupled data not detected by the automatic filtering.

- Evaluation and adjustment of the data processing based on preliminary inversion results.

Furthermore GPS data are lag-corrected to geographical positioned data/models at center between transmitter and receiver coils. The data uncertainty consists of a minimum of 3% as uniform data standard deviation (STD) plus the STD calculated from the data stacking. Averaged data resulting with STD over 30% are discarded from inversion.

## 2.4 Inversion

The electrical resistivities of the underground are then estimated using a series of 3D constrained 1D-inversions. The 1D

inversion is based on the *AarhusInv* code (Auken et al., 2015; Kirkegaard et al., 2015). This code is an implementation of a 1D non-linear damped least-squares solution, with a modeled transfer function for the TEM instrumentation. This function takes into account the transmitter waveform, the instrument low pass filters, the receiver bandwidth, the system geometry, the gate widths and the instrument front gate. However, in such an standalone 1D inversion, each model is totally independent of the neighboring ones. To account for the lateral continuity expected in geological environments, the spatially constrained inversion

(SCI) (Viezzoli et al., 2008) method was used. It applies 3D constraints to 1D inversion models both along and across the mapping lines, with a weight that is decreasing with distance. All the inversions were carried out with the *Aarhus Workbench* software.

The SCI inversion can be used with two different schemes of regularization: smooth or sharp. The smooth scheme tends to minimize abrupt changes in resistivity, in the vertical and horizontal directions. On the other hand, the sharp regularization





**Table 2.** Settings used for the model setup, the smooth and the sharp regularization.

| Parameter | Value |
|---|---|
| Model Setup | |
| Number of layers | 30 |
| Model resistivity start value (uniform - no prior) | 40 ohmm |
| Thickness of first layer (m) | 1 m |
| Depth to last layer (m) | 120 m depth |
| Thickness of layers | Log increasing with depth |
| Smooth Constraints | |
| Factor of horizontal contrains on resistivites | 1.5 |
| Reference distance | 10 m |
| SCI Constraints with distance | $1/distance^{0.75}$ |
| Prior, thickness | Fixed |
| Prior, resistivities | None |
| Minimum number of gates per inversion point | 2 |
| Sharp Constraints | |
| Factor of horizontal contrains on resistivites | 1.12 |
| Factor of vertical contrains on resistivites | 1.08 |
| Reference distance | 10 m |
| SCI Constraints with distance | $1/distance^{0.75}$ |
| Prior, thickness | Fixed |
| Prior, resistivities | None |
| Minimum number of gates per inversion point | 2 |
| Sharp vertical constrains | 500 |
| Sharp horizontal constrains | 300 |

scheme tends to minimize the number of resistivity changes, but will consequently result in more abrupt resistivity transitions and a potential more blocky model appearance. Both regularizations were used, and are included in the output data.

For each resistivity model, we estimate the depth of investigation (DOI) using a method based on the Jacobian Sensitivity matrix (Christiansen and Auken, 2012). This method has the advantage of taking into account the full transfer function, including system geometry, data uncertainty and the resistivity model. Two DOI thresholds values in the sensitivity matrix were used

to provide the reported DOI-standard, and the DOI-conservative values. As a guideline, the resistivity structures above the DOI conservative value are strongly data driven, while resistivity structures below the DOI standard value are weakly represented in the data. Normally one would blank the resistivity models below DOI standard value.

Inversion setup for the smooth and sharp inversions are summarized in the table 2.

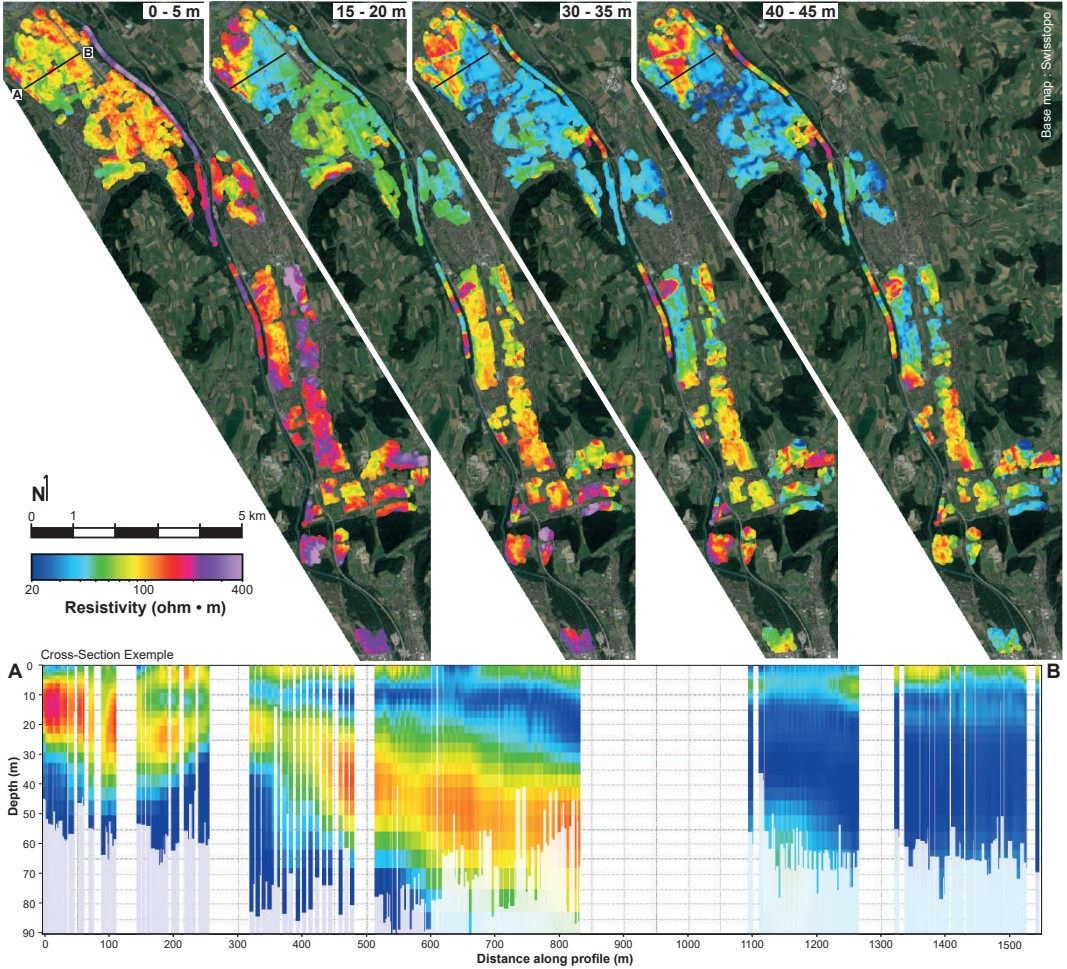

**Figure 4.** Mean resistivity maps at different depth intervals from the smooth regularization model. In addition, a cross section is included. Base map from Swiss Federal Topographic Office

The Fig. 4 presents some data extracted from the smooth regularization inversion. The spatial variations of the Quaternary
deposits, in both depth intervals and cross-section are clearly visible. Such variations in resistivities also indicates variations in lithologies, and therefore variations in hydrological proprieties.

## 3  Data Validation

$$\text{Data Misfit} = \sqrt{\frac{1}{N} \sum_{i=1}^{N} \frac{(d_{obs,i} - d_{frw,i})^2}{\sigma_{d,i}^2}} \tag{1}$$

where $d_{obs}$ is the observed data, $d_{fwr}$ is the forward data, $\sigma_d$ is the uncertainty of the observed data and $N$ is the total number
of data point. A data residual below 1 indicates that our final model response is within one standard deviation of the data, when

a value above 1 indicates a response out of one standard deviation. Figure 5b shows, a single data curve (error bars) and the forward response (line) from the resistivity model in Fig. 5a. The associated data-misfit for this model is 0.27. The data misfit for the all smooth inversion models is plotted in Fig. 6. As seen in Fig. 6, the data misfit is in general well below one and fully acceptable. 95% of the data is within 1 standard deviation, with a global misfit average of 0.65 and 0.52 respectively for the
sharp and smooth models.

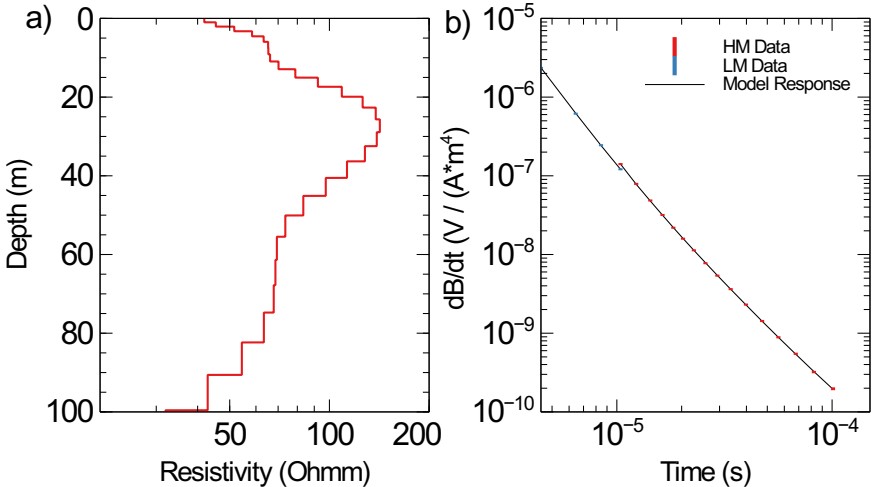

**Figure 5.** Example of one 1D smooth model at a location. Number 20350 at position 384744.1875/5196856 UTM 32 N. a) Smooth resistivity model b) its associated forward response in black line, with the LM & HM data point with red & blue error bars. The normalized data fit (see text) for this model/data curve is 0.27.

A manual inspection of the high data misfit models revealed that they are all associated to highly resistive models, and/or are close to man made electromagnetic noise such as roads, fences, or train tracks. A good example is the extreme south of the acquisition, that is one of the most resistive areas. This situation logically leads to a lower signal to noise ratio, and due to the spatial constraints of the inversion, it will consequently leads to an higher data misfit. However, they are usually restricted to
only a few local data-points, and the models are similar to neighbouring ones that has acceptable misfit. We therefore decided to keep them in the dataset.

Finally, users of the data should be aware that the footprint of the equipment is at least 9m at the surface (size of the equipment) and is increasing with depth and wave diffusion. Consequently, a sharp vertical transition in the geology for example, will tend to appear oblique in the resistivity data due to this effect. The resistivity models proposed here are only the
one that fits the best our data. However, the processed but uninverted data are provided, in order for others to perform inversions with different parameters or totally new inversion techniques.



**Figure 6.** Data Misfit over the acquisition area. Base map from Swiss Federal Topographic Office.





## 4 Code and data availability

After the data processing and the inversion, the uninverted processed data, the resistivity models and the associated forward responses from the smooth and sharp inversions had been produced. These data (Neven et al., 2020) are provided
in column based ASCII files. Each file structure is outlined in the following subsections. The can be freely downloaded on the Zenodo platform (https://doi.org/10.5281/ZENODO.4269887). The forward code used for the tTEM data presented here is the *AarhusInv* developped by the Aarhus University Hydrogeophysics group (Auken et al., 2015; Kirkegaard et al., 2015). The *AarhusInv* code is free to use for research purpose, and can then be combined with any other inversion technique (https://hgg.au.dk/software/aarhusinv/).

### 4.1 Processed data file

The `Processed_Data.dat` file contains the processed tTEM data and data uncertainties. Each line in the file corresponds to a low moment (LM) or high moment (HM) data stack for a given location. The RECORD number links the LM and HM data to a given resistivity model in the `*.inv` files. Number 9999 marks discarded data points or data points not present for the given moment. If all the data points of LM or HM are discarded then the data line is not present in the file. Gate center time
and other info is stated in the header lines. The data uncertainty is given as relative in log space. The upper and lower bounds of the data are then defined as :

$$\text{unc}_{\text{down}} = \frac{\text{DATA}}{1 + \text{DATASTD}} \tag{2}$$

$$\text{unc}_{\text{up}} = \text{DATA} \times (1 + \text{DATASTD}) \tag{3}$$

with $\text{unc}_{\text{down}}$ and $\text{unc}_{\text{up}}$ being the absolute lower and upper uncertainties, DATA the processed z-component dB/dt data value
and DATASTD the relative uncertainty. The structure is outlined in the following table 3.

### 4.2 Inversion Model File

The `Sharp_Model.inv` and `Smooth_Model.inv` files contain the resistivity models (layer resistivity and layer thicknesses). Each line hold a 30-layers resistivity model. The RECORD links the model to the data in the process data and forward data files. The file also contains the DOI, and the data fit. Note that the last layer (layer 30) does not have a thickness since
it continues to infinite depth in the modeling. Normally, the DOI standard values are used to blank the models in depths. The detailed file structure is provided in table 4.

### 4.3 Synthetic response file

The `Forward_Data_Sharp.dat` and `Forward_Data_Smooth.dat` files contains the forward responses of the sharp and smooth resistivity models. The structure of the forward data files is the same as the `Processed_Data.dat` file except
that the forward responses does not have associated data uncertainties. Detailed file structure is provided in table 5. The forward response was produced with the TEM 1D AarhusInv code.



**Table 3.** Structure of the .dat data file

| Processed_Data.dat | | | |
|---|---|---|---|
| Column | Label | Unit | Description |
| 1 | RECORD | | Global record number. Links the data to the resistivity model in the `*.inv` files |
| 2 | LINE_NO | | Line number (Line number 0 = data/model not tacked with a line number) |
| 3 | UTMX | (m) | UTMX coordinate, WGS 84 UTM zone 32N (epsg:32632) |
| 4 | UTMY | (m) | UTMY coordinate, WGS 84 UTM zone 32N (epsg:32632) |
| 5 | ELEVATION | (m) | Surface elevation |
| 6 | NUMDATA | | Number of data points (gates) in-use for the segment/sounding |
| 7 | SEGMENT | | Transmitter moment indicator. 1=Low moment, 2=High moment |
| 8-37 | DATA_# | (V/(Am4)) | Processed z-component dB/dt data value for gate number #. 9999 values = data not in-use/not present |
| 38-66 | DATASTD_# | STD | Data uncertainty for DATA_#, stated as a relative STD in log space. |

**Table 4.** Structure of the *.inv datafile

| Smooth_Model.inv, Sharp_Model.inv | | | |
|---|---|---|---|
| **Colum** | **Label** | **Unit** | **Description** |
| 1 | RECORD | | Global record number. Links the model the data in the *.inv files |
| 2 | LINE_NO | | Line number (Line number 0 = data/model not tacked with a line number) |
| 3 | UTMX | (m) | UTMX coordinate, WGS 84 UTM zone 32N (epsg:32632) |
| 4 | UTMY | (m) | UTMY coordinate, WGS 84 UTM zone 32N (epsg:32632) |
| 5 | ELEVATION | (m) | Surface elevation |
| 6 | DATAFIT | | Data fit (Data residual) |
| 7-36 | RHO_I_# | (Ohmm) | Resistivity of layer#. |
| 37-65 | THK_# | (m) | Thickness of layer #. |
| 66 | DOI_CONSERVATIVE | (m) | Estimated depth of investigation, conservative threshold value used |
| 67 | DOI_STANDARD | (m) | Estimated depth of investigation, standard threshold value used |

## 5   Conclusions

In this work, we presented the acquisition of more than 1400 hectares of tTEM measurement in the Upper Aare Valley, Switzerland. The dataset presented here is unique in terms of coverage, and combines it with a high spatial resolution. It reveals
the presence of various so far unexplained geological structures, in the shallow and in the deeper depths. This data set could be used for a wide range of different applications, including methodological developments as well as a better understanding of fluvio-glacial aquifers that represent a major source of drinking water supply in many places in the world.



**Table 5.** Structure of the *.syn data file

| Column | Label | Unit | Description |
|---|---|---|---|
| \multicolumn{4}{c}{**Forward_Data_Smooth.dat, Forward_Data_Sharp.dat**} | | | |
| 1 | RECORD | | Global record number. Links the data to the resistivity model in the `*.inv` files |
| 2 | LINE_NO | | Line number (Line number 0 = data/model not tacked with a line number) |
| 3 | UTMX | (m) | UTMX coordinate, WGS 84 UTM zone 32N (epsg:32632) |
| 4 | UTMY | (m) | UTMY coordinate, WGS 84 UTM zone 32N (epsg:32632) |
| 5 | ELEVATION | (m) | Surface elevation |
| 6 | NUMDATA | | Number of data points (gates) in-use for the segment/sounding |
| 7 | SEGMENT | | Transmitter moment indicator. 1=Low moment, 2=High moment |
| 8-37 | DATA_# | (V/(Am4)) | Model forward response, dB/dt, for gate number #. 9999 values = data not in-use/not present |

*Author contributions.* A.N. coordinated, conducted and supervised the field work. He performed the data analysis and inversion, prepared the data and wrote the paper. A.V.C. and P.M provided the instruments and software. They participated in the design of the measurements and checked the quality of data treatment and inversion. They edited and corrected the manuscript. P.R. obtained the funding for the survey. He supervised the work, participated to the field acquisition, and was involved in the data preparation, writing, and editing of the paper.

*Competing interests.* The authors declare no conflicts of interests.

*Acknowledgements.* The data set described in this paper was acquired within the framework of the Phenix project funded by the Swiss National Science foundation under the grant number 182600. The authors are thankful to all the people who contributed to the data acquisition and its inversion, and in particular: Rune Kraghede, Jesper Bjergsted Pedersen, Nikolaj Foged, Lucile Chauveau, Ilias Ben Ammar, Cyprien Louis as well as the local authorities and numerous farmers who provided access to their fields for the survey.



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
