# Peer review of "tTEM20AAR: a benchmark geophysical dataset for unconsolidated fluvio-glacial sediments"

_Earth System Science Data, 2020_

## Author Response (AR1)

**Response to the Referee Comments on "tTEM20AAR: a benchmark geophysical dataset for unconsolidated fluvio-glacial sediments"**

We again would like to thank all referees for their careful reading of our manuscript. We made an effort to address all comments that were made, and we believe that their suggestions improved the quality and readability of the work.

| Comment | Answer by the authors |
|---|---|
| Referee 1 : "details about the method used to detect coupled structures (line 102) should be added" | We added a short description to the paper. The method used is simply a slope filter. If an unrealistic gradient between two neighboring points is detected, these are considered suspicious and marked as coupled. Of course, we manually validate all of these. |
| Referee 1 : "In the conclusion, the authors mention that various so far unexplained geological structures are revealed by the new data, and I think that the paper would greatly benefit if 1-2 examples could be given." | We added a brief list of examples of findings in the text. However, the interpretation of these structures still requires a detailed sedimentological analysis as well as correlation with outcrops and boreholes. This work is currently in progress. It's too early to provide reliable conclusions now. Therefore we prefer to remain rather general for this paper. |
| Referee 1 : "but section 3 (Data Validation) starts directly with an equation, and a sentence should be added first to introduce the topic." | We added a short introduction |
| Referee 1 : "for instance, Figure 4 should also include the cross-section with sharpness constraint inversion. Figure 5 should also show the results of a 1D inversion with sharpness constraint applied. A noisy sounding should also be shown, for the reader to appreciate the overall quality of the data."

Referee 2 : "It would be interesting to see in figure 5 and/or 6, some sort of comparison of the two inversion results with sharp and smooth models as well between the conservative and standard DOI." | We modified Figure 4 to add the sharp cross-section, and include both DOIs for comparison.
Figure 5 now includes a noisy sounding and the two regularizations. |
| Referee 1 : "Around line 45, the authors should add that the dataset could also be used in future studies where other geophysical methods are used, to complement the analysis by performing joint inversion." | It was added. |

| | |
|---|---|
| Referee 2 : "I understand the with TEM data, there is obviously a DOI below which the resolution is very poor (as discussed here) but the is also close to the surface a depth above which the resolution is also very poor. Did you look at that in this dataset ?" | Yes ! There is indeed a depth above which the TEM is poorly sensitive. For the acquisition parameters we used, this depth is about 2 to 3 meters. We added a note in the text about this matter. |
| Referee 2 : "Do you consider the turn-off time to be constant ? Could it not change, even very little, when the properties of the very near surface change ?" | The turn-off time of the primary field induction loop is controlled by the transmitter current and the temperature of the electronics. It doesn't depend on the near-surface change. The system is water-cooled and regulated within a temperature limit (below 45 Celsius) so that the turn-off time doesn't change or so minimum that it does not have any influence on early time gates. This matter is also explained in the reference tTEM paper (Auken, et al. 2018) in more detail. |
| Referee 1 : "incomplete citation at line 69"
Referee 2 : "L19 " conduct to " should be "lead to"
Table 1 Precise in the first line "Tx No of turn" to prevent any uncertainty.
L69 incomplete citation
L66 "high efficient data collection" should be changed to either highly efficient or high efficiency data collection
L69-70 For consistency, choose "tTEM system" or "tTEM-system" and use it everywhere
L130-132 Again with "DOI-standard" or "DOI standard"
L131 weakly instead of weekly" | All the typos were corrected. |